# On Males, Antioxidants and Infertility (MOXI): Certitudes, Uncertainties and Trends

**DOI:** 10.3390/antiox12081626

**Published:** 2023-08-17

**Authors:** Manuel Alfaro Gómez, María del Rocío Fernández-Santos, Alejandro Jurado-Campos, Pedro Javier Soria-Meneses, Vidal Montoro Angulo, Ana Josefa Soler, José Julián Garde, Virginia Rodríguez-Robledo

**Affiliations:** 1Facultad de Farmacia, Universidad de Castilla la Mancha, 02071 Albacete, Spain; manuel.alfaro3@alu.uclm.es; 2SaBio IREC (CSIC—UCLM—JCCM), Campus Universitario, 02071 Albacete, Spain; alejandro.jurado@uclm.es (A.J.-C.); pedrojavier.soria@uclm.es (P.J.S.-M.); vidal.montoro@uclm.es (V.M.A.); anajosefa.soler@uclm.es (A.J.S.); julian.garde@uclm.es (J.J.G.)

**Keywords:** male infertility, idiopathic, oxidative stress, reductive stress, inflammation, biomarkers, antioxidants

## Abstract

Male infertility (MI) involves various endogenous and exogenous facts. These include oxidative stress (OS), which is known to alter several physiological pathways and it is estimated to be present at high levels in up to 80% of infertile men. That is why since the late 20th century, the relationship between OS and MI has been widely studied. New terms have emerged, such as Male Oxidative Stress Infertility (MOSI), which is proposed as a new category to define infertile men with high OS levels. Another important term is MOXI: Male, Antioxidants, and Infertility. This term refers to the hypothesis that antioxidants could improve male fertility without the use of assisted reproductive technology. However, there are no evidence-based antioxidant treatments that directly improve seminal parameters or birth ratio. In this regard, there is controversy about their use. While certain scientists argue against their use due to the lack of results, others support this use because of their safety profile and low price. Some uncertainties related to the use of antioxidants for treating MI are their questionable efficacy or the difficulties in knowing their correct dosage. In addition, the lack of quality methods for OS detection can lead to excessive antioxidant supplementation, resulting in “reductive stress”. Another important problem is that, although the inflammatory process is interdependent and closely linked to OS, it is usually ignored. To solve these uncertainties, new trends have recently emerged. These include the use of molecules with anti-inflammatory and antioxidant potential, which are also able to specifically target the reproductive tissue; as well as the use of new methods that allow for reliable quantification of OS and a quality diagnosis. This review aims to elucidate the main uncertainties about MOXI and to outline the latest trends in research to develop effective therapies with clinically relevant outcomes.

## 1. Introduction

Sterility is a disorder of the reproductive system that consists of the inability to achieve pregnancy after 12 months of sexual intercourse without the use of contraceptive methods. It is estimated to affect between 8 and 12% of reproductive-aged couples worldwide [1]. Infertility refers to the problem of couples who achieve a pregnancy but do not carry it to term. Male infertility (MI) is defined as the inability of a male to make a fertile female pregnant, also for a minimum of a one year of unprotected intercourse [2]. As for the male factor, males are found to be solely responsible for 20–30% of infertility cases and contribute to 50% of cases overall [3]. A multitude of causes and risk factors contribute to MI, which could be classified as congenital factors (anorchia, anatomical obstructions, congenital absence of vas deferens, and cryptorchidism), genetical factors (Y chromosome microdeletions, chromosomal abnormalities, and genetic endocrinopathy), acquired factors such as infections, testicular traumatism or torsion, varicocele, hormonal disorders, immune infertility due to antisperm antibodies, postinflamatory conditions (epididymitis), germ cell tumors, or sexual dysfunction; lifestyle factors, which include smoking, obesity, alcohol intake, psychological stress or recreational drugs, and exogenous factors like chemotherapy, radiation, heat, or medications [4]. However, despite the advances in the understanding of MI, the etiology is unknown in 30–50% of patients [5]. This type of infertility, the origin of which is unknown, is classified as idiopathic male infertility (IMI) or unexplained male infertility (UMI). UMI is defined as infertility of unknown origin with completely normal findings at semen analysis. In this kind of infertility, female factors associated with their partner has also been ruled out. In contrast, men with IMI are those who have an unexplained decrease in semen quality with no history of fertility problems and normal physical examination and endocrine laboratory tests. Their routine semen analysis shows a decrease in sperm count (oligozoospermia), a decrease in motility (asthenozoospermia), or an increase in the proportion of abnormal forms (teratozoospermia) [5,6].

A proper diagnosis of male infertility requires a complete anamnesis and evaluation of the medical history including data about sexual habits, previous fertility, current or past diseases, and exposure to medications and toxic substances. Physical examination is also a key element in the diagnosis of male infertility, including an assessment of body habitus, secondary sexual characteristics, and genitalia. Hormonal evaluation is another important tool for achieving a successful diagnosis and should include analysis of follicle-stimulating hormone and total testosterone [4]. But semen analysis is probably the best diagnostic tool. The World Health Organization (WHO) recommends this analysis as the first step in the evaluation of male fertility potential. Conventional analysis follows the WHO laboratory manual for the examination and processing of human semen, sixth edition. 2021. This evaluation includes parameters as pH, liquefaction, appearance, semen volume, sperm concentration, morphology, and viability. Some complementary tests could be useful, such as genetic tests, immunoassays, post-coital test, testicular biopsy, and imaging techniques [2]. Nevertheless, new methods and biochemical approaches are needed to improve the diagnosis, tracking and treatment of male infertility [7].

As mentioned above, MI depends on a wide range of factors, so treatments vary according to the underlying disease and the degree of the impairment of the male’s fertility [8]. Treatments include hormonal therapy with LH, gonadotropin or anti-estrogens such as clomiphene citrate (for endocrine disorders); pharmacological treatment using drugs like bromocriptine or systemic corticosteroids; antioxidant supplementation; surgery (for disorders such as varicocele or epididymal obstructions) and assisted reproduction techniques (ART), such as IVF, ICSI or IUI [8,9,10]. 

However, despite significant advances in the diagnosis and management of male infertility, there are no evidence-based treatment guidelines available for idiopathic MI. A fact which highlights this is the influence of oxidative stress (OS), which plays an independent role in the etiology of MI, with 30% to 80% of infertile men having elevated seminal reactive oxygen species (ROS) and nitrogen species (RNS) levels [6,11]. In this regard, it is worth mentioning that the sperm cells are highly sensitive to oxidative damage and they are incapable of repairing damage by oxidative stress because they suffer from a lack of essential cytoplasmic enzymes. Therefore, based on the understanding that antioxidant deficiency exacerbates this oxidative imbalance, antioxidant supplementation represents a prime area of therapeutic research [12].

This aspect will be discussed extensively during the review, considering the wide and varied bibliography published so far. In this review paper, we will appraise the certainties that are supported by years of study, expose the uncertainties raised in studies under different approaches, and present the latest trends in improving the diagnosis and treatment of MI.

## 2. Certitudes in Male Infertility

Since the 1970s, there has been an exponential growth in scientific publications related to “Male Infertility”. But it is from the year 2000 onwards that there was a clear upward trend in the number of articles including the term MI, exceeding 1000 articles published per year only utilizing a PubMed search. In 2021, a record number of 2881 articles were published, exceeding 2700 publications in PubMed in 2022 and more than 900 so far in 2023. Conducting a search on the same scientific platform including, in addition, the terms, “oxidative stress” and “antioxidants”, with a filter applied to only show articles published in 2022 and 2023, over 250 articles were found, of which, 50 are reviews. After reviewing all titles and abstracts, around 40–50 of them were been selected and included in this review article. Most of them share among their proposed studies and conclusions some certainties that will be deeply discussed below. Other official databases, such as Web of Knowledge and Google Scholar, were consulted, obtaining smaller number of articles using similar keywords and filters compared to PubMed.

### 2.1. Oxidative Stress and Idiopathic Male Infertility: MOSI

One of the main certainties of male-factor infertility is that it has become one of the major global health problems, currently accounting for 20–30% of infertility cases worldwide. As previously mentioned, idiopathic male infertility accounts for a large majority of cases where the underlying cause(s) remains elusive [13,14]. In this respect, most of the articles published over the last 10 years show a strong correlation between elevated reactive oxygen species (ROS) levels in semen and worse reproductive outcomes. Therefore, another certainty is that oxidative stress (OS) is currently the most widely accepted mechanism as a key factor in explaining idiopathic MI cases, [15] through phenomena such as mitochondrial dysfunction, lipid peroxidation, DNA damage and fragmentation, and finally, sperm apoptosis [16,17]. Thus, all factors which have been traditionally associated with male infertility and poor sperm quality, such as alcohol, smoking [18], obesity, varicocele, infections, and psychological stress may indeed exploit their effect through oxidative stress [19]. With regard to the latter and having in mind the strong relationship between OS and MI, some authors are currently starting to refer to the category, MOSI (Male Oxidative Stress Infertility) [20], as a medical term to describe infertile men with abnormal sperm parameters and oxidative stress (OS), including those previously classified as having idiopathic infertility [21].

One of the first authors to propose the term MOSI was Agarwal et al. at 2019, as a novel descriptor for infertile men with abnormal semen characteristics and OS, including many patients who were previously classified as having idiopathic male infertility [5] (see Figure 1). Miccogullari et al. have published a study on the role of thiol/disulfide homeostasis with a novel and automated assay in MOSI [22], for the determination of oxidant/antioxidant status in serum samples by using a novel test of patients with IMI and comparing their results to those of healthy controls.

Therefore, since OS occurs when seminal reactive oxygen species (ROS) generation exceeds endogenous antioxidant capacity, oral antioxidant supplementation has been proposed, from many years ago [23] until now [24], as a key strategy in the treatment of IMI. Another certainty is, consequently, the popularization in the scientific field of the use of antioxidants during the last decades, despite the lack of conclusive results regarding their use. This may be due to two main reasons. The first is that these compounds are widely available and inexpensive when compared to other fertility treatments and also because of the proved safety of oral supplementation with antioxidants [23], being an attractive area for the study of MI [25].

### 2.2. Antioxidants, as an Alternative in the Treatment of Idiopathic MI: MOXI

In this section, we will discuss in detail the well-known antioxidants that have been traditionally proposed as possible treatment for IMI, since they are the first line of defense against MOSI [26,27]. But before that, we must classify antioxidants in two different types: enzymatic and non-enzymatic. The enzymatic antioxidant system in semen is composed of endogenous molecules, the three most relevant ones being superoxide dismutase (SOD), catalase (CAT), and glutathione peroxidase (GPX) [28]. However, non-enzymatic antioxidants can be both endogenous or exogenous, and the most important ones in spermatozoa are glutathione, selenium, carotenoids, such as lycopene, ascorbic acid, and -tocopherol, which exert their antioxidant effects through direct neutralization, promoting expression of antioxidant enzymes, or acting as cofactors for antioxidant enzymes. There are many papers and review articles on studies of antioxidants’ effects on MI, particularly the review published in 2010 by Agarwal et al. [27], based on many reputable studies, which described the effects of various antioxidants such as carnitines, vitamin C, and vitamin E as a first-line treatment, and others such as as glutathione, selenium, and coenzyme Q10 as the second line.

Years later, Adewoyin et al. published a review about the effect on seminal OS of natural antioxidants and phytocompounds [28]. In this paper, it was found that vitamins E and C, carotenoids and carnitine were notably beneficial in restoring a balance between ROS generation and scavenging activities. These reviews prove that, for years, studies have been carried out using conventional antioxidant compounds individually or in combination, observing that, in general, these caused positive effects in the treatment of MI.

However, it was not until the year 2020 that the term, MOXI (Male, Antioxidants and Fertility) trial, was described by Steiner et al. [29]. This study was designed to test the hypothesis that antioxidants would improve male fertility without the use of assisted reproductive technology (ART) (Table 1). They examined the efficacy of a commercially available antioxidant combination pill, containing 500 mg of vitamin C, 400 mg of vitamin E, 0.20 mg of selenium, 1000 mg of L-carnitine, 20 mg of zinc, 1000 mg of folic acid, 10 mg of lycopene daily for three to six months, as a treatment for infertile men with abnormal semen parameters. The conclusion of this study suggested that the commercial antioxidant mixture did not improve semen parameters or DNA integrity among men with male factor infertility, nor did it improve in vivo pregnancy or live birth rates. This work has been included as a reference in other similar studies. For example, Knudtson et al. [30] studied the role of plasma antioxidant levels in male fertility in the USA (α-tocopherol, zinc, and selenium) through analysis of the MOXI randomized clinical trial. In this study, the primary endpoints in this secondary analysis were semen parameters, DNA fragmentation, and clinical outcomes including pregnancy and live birth. There was no association between selenium, zinc, or vitamin E levels and semen parameters or DNA fragmentation, and no association was observed between antioxidant administration and semen parameters or clinical outcomes in male infertility in men with sufficient or higher plasma antioxidant levels. So, it does not appear to confer benefits on semen parameters or male fertility.

Finally, a review paper has recently been published by Chen et al. [25], including several studies that are less susceptible to bias because of being placebo-controlled and more highly powered. Among them, the MOXI trial is of particular interest, being a large, multicenter RCT, which demonstrated no improvement in semen parameters or live-birth rates with antioxidant use [29,30]. In this regard, the review authors proposed to stop recommending antioxidants for IMI treatment given the recent studies that suggest a lack of efficacy, or else continue using them until overwhelming evidence is found, due to their relative safety, and the ease of being commercially available and inexpensive as we discussed above lines.

In this way, a systematic review and meta-analysis of randomized controlled trials (RCT) published in early 2023 by Agarwal et al. [31] assesses the impact of antioxidant therapy on natural pregnancy outcomes and semen parameters in MI. Among more than 1.300 papers, 45 RCT, and 4332 infertile patients, a significantly higher pregnancy rate was found in patients treated with AOX compared to placebo-treated or untreated controls. However, no effects on live-birth or miscarriage rates were observed in four studies, because, according to the author of the review, very few RCTs specifically assess the impact of AOX. Therefore, further RCTs assessing the impact of AOX on live-birth rates, miscarriage rate, and SDF will be helpful.

Barati et al. [32] explain the roles of **oral antioxidants** and herbs in coping with oxidative stress in male infertility, among other studies, in a review published in 2020. Low-molecular-weight compounds later described and nutrients such as selenium, zinc, and copper can protect the body against OS. For years, many urologists have prescribed oral antioxidants for infertility and nowadays, the use of herbal therapy has also been considered to prevent infertility because these antioxidants can reduce the destructive effects of oxidative stress [33].

### 2.3. Other Antioxidants and Their Role as Biomarkers

Therefore, it seems timely and relevant to dedicate a section to explore evidence on other types of antioxidants, traditionally used or having emerged in recent years, taking into account recently published review papers.

Lucignani et al. recently published a review based on the evidence of the efficacy of coenzyme Q10 (CoQ10) and melatonin on male infertility [34]. On the one hand, CoQ10, is one of the antioxidants historically studied for years [35,36] and is a fat-soluble ubiquinone with intracellular antioxidant activity, the lack of which has been observed in various sub-fertile conditions, (e.g., varicocele, oligozoospermia) [37], while melatonin is a hormone secreted by the pineal gland of the brain, and its metabolites act as free radical scavengers, hence protecting cells from oxidative stress, which could have a significant impact on infertility [38]. In both cases, the conclusion is that larger studies are needed to assess their clinical efficacy and potential in male infertility.

N-Acetyl-l-cysteine (NAC) is a reduced glutathione precursor with strong anti-inflammatory and antioxidant effects. It is widely used in the respiratory system, treatment of diseases of the digestive and cardiovascular systems [39]. In addition, some articles have been currently published on the role of NAC orally in IMI [40]. Daily N-acetyl-cysteine administration resulted in a statistically significant improvement in semen parameters, particularly sperm motility and normal morphology, but had no effect on the patient’s hormone profile [41,42]. Its potential therapeutic benefits lie in its direct antioxidant capacity, thanks to its ability to chelate heavy metals and the reducing effect of the thiol group (-SH) present in its molecular structure, and indirectly, as a precursor of glutathione, as mentioned above. It also has anti-inflammatory and immunomodulatory effects, which makes it an interesting alternative for the treatment of IMI [43].

Antioxidant compounds may also have a dual role, not only as bioactive molecules that can be used for the treatment of IMI, but also as biomarkers. The identification and then determination of certain biomarkers has great potential for the diagnosis, confirmation and prevention of imbalances affecting, in this case, fertility. In addition, they can be used to determine the physiological state of an organism or to measure the response to a particular drug. By evaluating these molecules, it is possible to get closer to the possible background of the disorder causing a fertility problem.

Several studies have been carried out in this regard, among which the determination of levels of coenzyme-Q10 and α-tocopherol (Vitamin E) in blood plasma [44] and seminal fluid, are proposed as important metabolic biomarkers for diagnosis, treatment, and even monitoring of male infertility by measuring of concentration levels of bioactive compounds [45]. After administration during 3 and 6 months of treatment, concentration levels of antioxidants are significantly increased and the OS decreased. On the other hand, Vitamin D has been also studied as a potential biomarker of MI [46] and a review article with this issue has been published. Such article even explores the possible relationship between obesity and vitamin D deficiency, and its role in male infertility has been also published [47]. Other biomarkers have been described in the literature as good candidates for the diagnosis of MI, such as 17-hydroxyprogesterone (17-OHP) and insulin-like factor 3 (INSL3) as accurate secondary biomarkers of intratesticular testosterone in the context of male infertility [48].

Finally, it should be noted that currently, -omics platforms (genomics, epigenomics, transcriptomics, proteomics and metabolomics, among others) are very powerful tools to identify the most robust molecular biomarkers for the diagnosis of diseases, in this case, in sperm and seminal plasma. For this reason, these techniques may be interesting for the diagnosis of male infertility, and to evaluate the clinical uses of some biomarkers [49]. Proteomics untargeted approaches, depending on the study design, might provide a plethora of biomarkers not only for a male infertility diagnosis but also to address a new MS-biomarkers classification of infertility subtypes [50]. On the other hand, seminal fluid contains the highest concentration of molecules from the male reproductive glands; therefore, these molecules could be potential seminal biomarkers in certain male infertility scenarios, including natural fertility, differentiating azoospermia etiologies, and predicting assisted reproductive technique success [51]. Seminal protein-based assays of TEX101, ECM1, and ACRV1 are already available or under final development for clinical use.

## 3. Uncertainties in MI

The review papers published during the last 5 years incorporate a series of conclusions that bring some uncertainty to the years of study on the use and effect of antioxidants on IMI. In this section, we will review the most current studies, paying special attention to the results obtained and the advances that the conclusions provide us with.

In this sense, Agarwal et al., in a review published in 2019, noted that current treatments for OS, including the use of antioxidants, are not evidence-based and have the potential for complications and increased healthcare-related expenditures [5]. In addition, the measure of reductants (antioxidants) and oxidants levels in biological fluids [52] could become an adjunct component of semen analysis due to its robust association with impaired sperm function. Years later, in the same way, Dutta et al. [24] indicated that although several antioxidants reportedly have a positive impact on fertility parameters, further randomized controlled trials are needed to demonstrate the efficacy and safety of antioxidant supplements for treating IMI, as well as to find the optimal dose of each antioxidant to improve fertility parameters [27]. This same author explains that in the absence of quality detection methods for the determination of OS biomarkers in MI, the use of antioxidants may be arbitrary or the dosage and frequency may be inadequate, causing the effect known as “reductive stress” [24]. Other review article explains the redox regulation in male reproduction, analyzing the impact of both the OS and reductive stress on sperm function [21]. In addition, the inflammation is closely linked to OS, resulting in an exaggeration of cellular damage and disruption of male reproductive tissues. Therefore, the overuse of antioxidant supplements, leads to the so-called “antioxidant paradox” a term coined in the year 2000 by Halliwell [53]. This effect is based on the importance of maintaining a delicate cellular redox balance to support physiological levels of ROS production for robust male reproductive functions [54] and defines the unexpected effects of antioxidants overuse. In the same way, years later Henkel et al. explained how the over-usage of antioxidants may cause reductive stress, and manifest contradicting impacts of antioxidant treatment on male reproduction [55]. This article provides an excellent and unique explication of the vicious cycle of OS and inflammation (Figure 2) [24] in addressing the common failure of antioxidant treatment for male infertility. Therefore, given the reasons antioxidants have not provided obvious beneficial effects, several explanations have been proposed [53,56].

Among the hypotheses that have been considered, one of those mentioned above is the related to the type and dose of antioxidants applied in clinical trials. These may not have been specific enough to alleviate OS in a tissue- or cell-specific manner, or they either had no effect or were detrimental impacts. Another one is the selection of antioxidants that do not simultaneously inhibit both OS and inflammation, or the use of nonselective agents that block some oxidative and/or inflammatory pathways while exacerbating the effects of others [24]. Nevertheless, it would be necessary to quantify both the redox and the inflammatory status before, during, and after the antioxidant therapy applying an appropriate methods in order to establish the validity of the results in many clinical trials [57].

Some recently published reviews even go so far as to propose a new time to design novel strategies for antioxidant therapy [58] based on increasing the understanding of the pathophysiology of MI and on the particular biological properties of sperm and their relationships with OS. This review stresses that, since low levels of ROS are necessary during processes such as capacitation and acrosome reaction, the overuse of antioxidants could be deleterious due to reduced ROS generation and intracellular concentration [59]. Therefore, it is once again evident that there is currently no clear consensus on the optimal antioxidant constituents or regimen and neither applicable antioxidant treatment against this problem. Ávila et al. even warn that the antioxidants so widely used in clinical trials for male infertility such as vitamins C and E, -carotene, selenium and zinc [60], have shown detrimental effects in the context of fertility such as increased sperm decondensation, among other effects. However, this once again points to the importance of not only describing the optimal antioxidant doses but also to carrying out studies aimed at understanding the synergistic effects of antioxidant compounds [55].

In the same way—and continuing with the section dedicated to the uncertainties that some studies have incorporated in recent years regarding the use of antioxidants to improve semen parameters or directly influence the birth ratio—we are going to describe the most outstanding ones. Some recent studies using a commercially available mix of some of these antioxidants, such as multivitamins (vitamin C and vitamin E) selenium, L-carnitine, zinc, folic acid, and lycopene [29] have been discussed in the review currently published by Chen et al. [25]. The conclusions in all cases were that a mix of antioxidants neither improved semen parameters nor increased live-birth rate [30,61]. In fact, the recently published prospective study assessed the impact of 3 months of lifestyle changes in conjunction with oral antioxidants (multivitamins, CoQ, omega-3, and oligo-elements) and no differences in sperm parameters or in the treatment group were observed [61]. By this means, it seems that the *Mediterranean diet* might protect against MI compared to the *Western diet*, although the biochemical mechanisms regarding sperm quality are still poorly understood. However, a *vegetarian diet* has been shown to be a controversial factor for MI and other chronic diseases [62], in a review published at 2022 by Ferramosca et al. [63] about diet and antioxidants on sperm and energetic metabolism. This review aimed to analyze the molecular effects of single nutrients, such as saturated fatty acids (SFA) or long-chain polyunsaturated fatty acids (PUFA), on sperm quality parameters. The administration of PUFA, especially omega-3, led to an increase in mitochondrial energetic metabolism and a reduction in oxidative damage. On the other hand, while vegetables and fruits are rich in antioxidant molecules, which can act as sperm ROS regulators by reducing sperm DNA damage and by increasing sperm motility and vitality [63], some studies already indicate that the *vegetarian diet* reduced sperm concentration and motility [64]. This negative effect on sperm parameters could be attributed to estrogenic compounds or chemical residues in the diet, which had a negative effect on sperm parameters [65,66,67]. Other nutrients such as carbohydrates and proteins can be considered modulators of oxidative stress and testosterone levels, which are strictly linked to sperm mitochondrial function, a key element for sperm quality.

Ideally, an oral antioxidant should reach a high concentration in the reproductive tract and restore vital item for spermatogenesis [32]. However, the blood–testis barrier (BTB), produced by Sertoli cells (SCs) near the base of the epithelium of the seminiferous tubules, is one of the strongest blood–tissue barriers in mammals. Limiting the paracellular and transcellular diffusion of water, electrolytes, ions, hormones, paracrine factors, and other exogenous biomolecules provides a unique microenvironment for spermatogenesis and ensures the normal physiological function of the testicles. The germ cells of the BTB and testicles also express many drug transporters which actively pump drugs out of the testicles. Overall, the barrier function of the BTB combined with the drug transport network in the testicles prevents the delivery of drugs to them [68]. There are still many challenges for drugs that must cross the BTB to elicit therapeutic function in the tissues. For example, when making the BTB leakier for drug delivery and improving drug utilization, we must also protect the integrity of the BTB. If there is irreversible damage, it can seriously affect male reproductive function.

## 4. TRENDS around MOXI

Taking into account the certainties and important uncertainties that we have described throughout the review, it is more than pertinent to dedicate a section to comment on the trends highlighted in the most recent work cited above.

Continuing to the last section, future studies should compare and study biomolecules that can bypass the BTB and further determine whether there are specific chemical groups and characteristics that are essential to bypass the BTB. This will provide a foundation for the development of chemotherapeutics, antivirals, and male fertility drugs to reduce testicular tumor recurrence and viral infections in semen and improve male fertility. Moreover, we must consider whether men of different races and ages respond identically to the same therapeutic drug as well as control the cost of drugs, which are essential elements of the widespread use and promotion of therapeutics [68].

Several studies published on the treatment of idiopathic MI with antioxidant compounds (MOXI) have established future research avenues. This is mainly due to the large number of studies and papers that have been published over the years that have not yet provided conclusive results on the success (or failure) of MOXI.

One of the most widely accepted trends is for interventions based on certain antioxidant molecules (not arbitrary) or mixtures of them in controlled doses, which can be related to specific biochemical mechanisms and also contribute to the knowledge of these mechanisms, in order to improve treatment in the future. In this sense, and as it is before mentioned, Ávila et al. in their review published in 2022 [58] propose a novel antioxidant therapeutic strategy based on specific intracellular phenomena and sensitive sperm development timeframes to obtain positive achievements. As an example, the authors include three different antioxidants with suitable properties, aiming to improve the clinical outcomes of MI and taking in mind the time course of stages of spermatogenesis and OS. The proposed antioxidants are a long chain-3 polyunsaturated fatty acid for the mitotic phase of spermatogenesis, resveratrol because of its role as a direct and indirect antioxidant involved in meiosis, and melatonin in the spermiogenesis phase.

Our increasing physiological and biochemical knowledge of “the redox regulation in male reproduction” helps us to relate it to other processes such as inflammation, resulting in an exaggeration of cellular damage and disruption of male reproductive tissues. Therefore, the overuse of antioxidant supplements leads to the so-called “antioxidant paradox”, which is a term that was coined in the year 2000 by Halliwell [53]. Therefore, another trend that arises is the dual treatment of IMI, taking into account not only the OS but also the inflammatory process. When oxidative stress appears as a primary disorder, inflammation develops as a secondary disorder and further enhances oxidative stress. On the other hand, inflammation as a primary disorder can induce oxidative stress as a secondary disorder, which can further enhance inflammation [69]. It is important to underline that when chronic inflammation occurs, there is impaired accessory gland function, sperm transport obstruction, and spermatogenesis dysregulation, all of which may lead to poor semen quality [24]. Thus, drugs with dual antioxidant and anti-inflammatory potential, could achieve better results than drugs that only focus on an individual problem. An interesting example is NAC, a drug that has the antioxidant effect mentioned above but also inhibits NF-kb activation (nuclear factor kappa-light-chain-enhancer of activated B cells) and subsequent pro-inflammatory cytokine production, such as TNF-α, IL-1, and IL-6 [43].

In the same way, Ferramosca A. et al. [63] emphasize that understanding the biochemical mechanisms responsible for sperm quality will lead to more targeted and effective therapeutics for male infertility. Moreover, in this article, the authors indicate that an adequate intake of antioxidant molecules can be effective in the prevention and/or treatment of male infertility [65]. As discussed above, other authors such as Dutta et al. [24] have written about the “the antioxidant paradox”, in which not only inflammatory processes but also the effect known as “reductive stress” take center stage, which is why the use of antioxidants may be arbitrary or the dosage and frequency may be inadequate. Therefore, in this review, we aim to classify the studies not by the types of antioxidants used for the treatment of IMI but by considering those interventions that have been successful [31]. In this way, we could continue with lines of research that would advance applications with a certain success rate, and on the other hand, new research lines would be opened in which MOXI treatments with less success could be studied in other pathologies.

Other works point to new measurement tools for obtaining quality and comparable results, helping to draw conclusions that provide some certainty [24]. Specifically and in this way, Agarwal et al. [5] propose the measurement of the oxidation-reduction potential (ORP) by using an easy, reproducible, and cost-effective test as an useful clinical biomarker for the classification of MOSI in men with abnormal semen analysis and male infertility [70,71]. The ORP test may be used to measure the levels of reductants (antioxidants) and oxidants in a variety of biological fluids and may provide a reliable approach for administering antioxidant therapy while minimizing the risk of antioxidant overdose.

Along the same lines, recent works incorporate hybrid instrumental techniques for the determination of biomarkers traditionally described in the literature [72] for the measurement of OS by employing assays such as the ones that we can see in Table 2.

The analytical instrumentation provides high precision, accuracy, and more specificity. On the other hand, simultaneous determination of several analytes when separation techniques are included (mainly chromatography) in complex matrices as is the case of biological samples such as whole semen, seminal plasma, blood, tissues, etc. In addition, when separation techniques are coupled to a powerful detector (e.g mass spectrometry), chemical information on the analytes can be obtained, allowing a better correlation and improvement of the results because the methodologies are specifically developed for the quantification of these biomarkers. In this way, the determination of MDA and 4-hydroxy-2-nonenal as biomarkers of lipidic peroxidation by using diode-array [73] and fluorescence detection [74], or a mass spectrometry detector [75] using derivatization procedures have been published. Nevertheless, much more research is needed into the development of new analytical methodologies using coupling analytical techniques to replace the currently well-established assays used for the measurement of OS biomarkers. Other biomarkers such as 8-hidroxi-2’desoxiguanosine for DNA damage [76] or reduced and oxidized glutathione as have been also quantified via LC-MS/MS [77,78].

In the same way, and as mentioned above, another trend is towards using -omics platforms (genomics, epigenetics, proteomics, transcriptomics, and metabolomics) that allow us to identify new and more specific biomarkers that will enable us to expand and improve studies. Years ago, Bieniek et al. [51] advised that future ventures will need to find predictive biomarkers due to the poor diagnostic ability of current assays for male infertility. In this review, panels of DNA, RNA, proteins, or metabolites are proposed to understand the pathophysiologic processes of male infertility. However, the rapid expansion of mass spectrometry (MS) technology in the field of the ‘omics’ disciplines has incredibly proven the great potential of MS-based diagnostic tests to revolutionize the future of pathology, microbiology, and laboratory medicine, as Preianò et al. [50] claim in their recently published review. This article reports the most recent studies and the efforts of the scientific community to address proteomic research via untargeted approaches to detecting biomarkers of infertility with a special focus on experimental designs and strategies (bottom-up and top-down), as can be seen in Figure 3. However, the correct diagnosis of male infertility requires an accurate validation of putative signature molecules from the discovery phase as is addressed in only a few studies [79,80,81]. Once these hurdles are cleared, the future developments for laboratory medicine diagnosis focus on cheap MS instrumentation with ultra-fast high-throughput features as well as highly specialized and experienced personnel for the correct interpretation of the quality of the MS outputs.

Finally, another emerging line of research around MOXI is the use of **nanotechnology as a tool** to transport bioactive molecules, mainly those with stability problems, poor permeability to biological membranes, or toxicity [82,83]. For example, vitamin E is a well-known and powerful lipid-soluble antioxidant which is widely used to prevent oxidate stress. However, its administration is sometimes restricted due to the lack of stability and the toxicity of the organic solvents required for administration [84]. The use of vitamin E delivery systems such as nanoemulsions [85] or hydrogels [86] offers several advantages compared to conventional administration. Drug delivery systems (DDS), and more specifically, nanotechnology, provide, among other aspects, protection of the active molecule or selective interaction with target cells, resulting in more effective therapies. Along this line, the paper recently published by Jurado-Campos et al. reported lipid-based vitamin E nanodevices as a tool for preserving red deer post-thawed sperm samples against oxidative stress. This report was a pioneering work in assisted reproductive technologies in which vitamin E nanodevices were tested in ram spermatozoa, as a proof of concept.

## 5. Conclusions

ROS overproduction leads to OS, which plays a central role on the etiology of MI. Despite the increased research and use of antioxidants in recent years, antioxidant therapy has not presented solid benefits. For this reason, new terms such as “the antioxidant paradox” have emerged to explain this issue. This term, which is the most important one, is used to refer to problems such as the lack of robust methods for OS quantification and the lack of treatments simultaneously targeting OS and inflammation. Indeed, the uncontrolled use of antioxidants for treating MI may induce “reductive stress”. Such problems could explain why the results have not been as expected. However, given that molecules such as these have been widely studied and are affordable, there is now a need to look for new strategies that bring clinical relevance to the use of antioxidants. Among them, it is worth highlighting the use of nanotechnology as a tool that helps molecules to bypass the blood–testis barrier to achieve high concentration on testis and the screening of compounds with dual antioxidant and anti-inflammatory action. Finally, the incorporation of proteomics/metabolomics and hybrid instrumental techniques could provide great diagnostic potential. These strategies may lead to promising results for the treatment of MI.

## Figures and Tables

**Figure 1 antioxidants-12-01626-f001:**
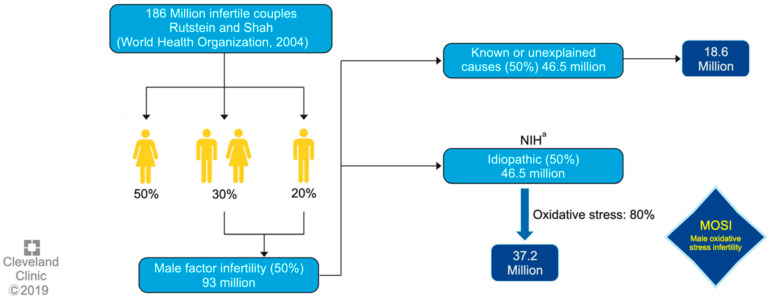
Worldwide incidence of MOSI in infertile men. ^a^ National Institutes of Health (NIH). Reproduced with permission from Figure 3, Agarwal, A., et al., World J Mens Health 2019 [5].

**Figure 2 antioxidants-12-01626-f002:**
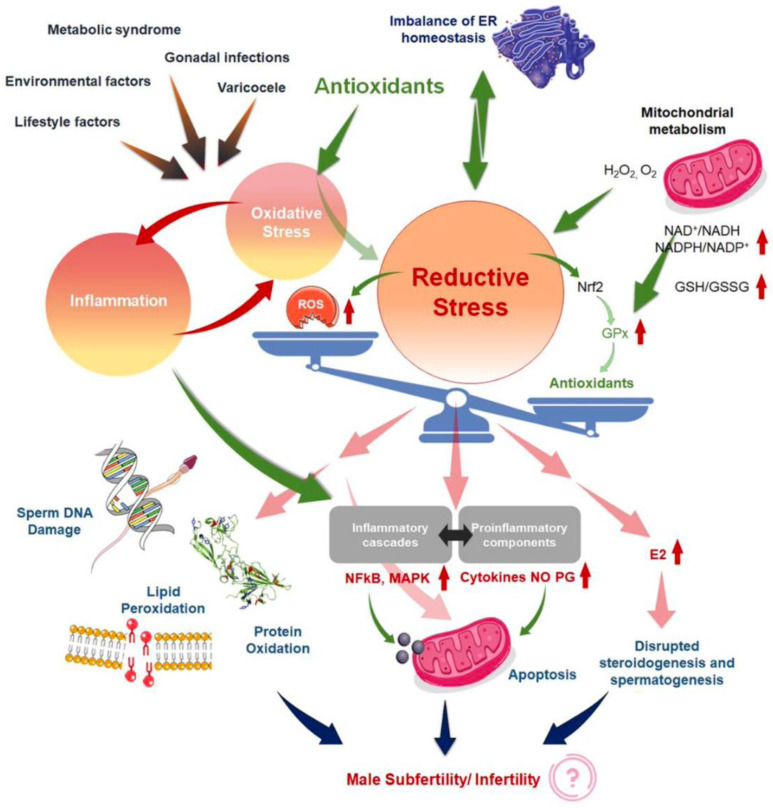
Mechanisms to explain ‘antioxidant paradox’ pertaining to male infertility, both by the induction of reductive stress and the failure to address the interconnected link of oxidative stress (OS). Reproduced with permission from Figure 1 of Dutta, S., Antioxidants 2022.

**Figure 3 antioxidants-12-01626-f003:**
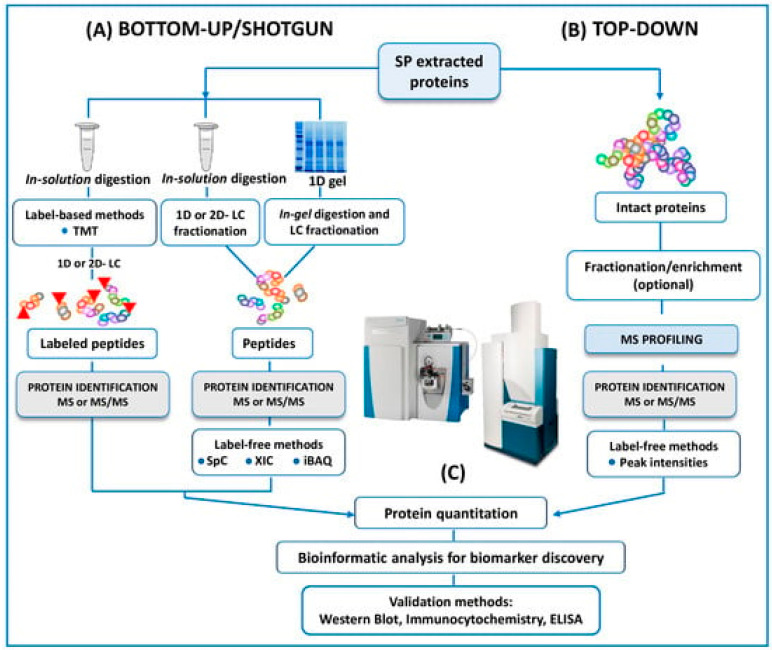
A schematic outline of all the protocols here reviewed in both bottom-up and top-down untargeted MS approaches for the proteomic analysis of human SP. Reproduced with permission from Preianò, M. et al., Int. J. Mol. Sci., 2023.

**Table 1 antioxidants-12-01626-t001:** Characteristics at screening for all enrolled men in the Males, Antioxidants, and Infertility (MOXI) randomized clinical trial. Reproduced with permission from Table 1 of Steiner, Antioxidants 2020.

Data Are Presented as the Number (%) or Median (Interquartile Range).		
	Antioxidants (*n* = 85)	Placebo (*n* = 86)
Age (years)	34.0	34.0
	(30.0, 37.0)	(30.0, 38.0)
Body mass index (kg/m^2^)	27.8	27.6
	(24.2, 31.7)	(24.4, 31.0)
	*n* = 82	
Ethnicity		
Hispanic or Latino	7 (8.2)	5 (5.8)
Non-Hispanic	72 (84.7)	78 (90.7)
Unknown	6 (7.1)	3 (3.5)
Race		
White	63 (74.1)	69 (80.2)
Black	6 (7.1)	7 (8.1)
Asian	7 (8.2)	2 (2.3)
American Indian or Alaska Native	1 (12)	1 (12)
Unknown	8 (9.4)	5 (5.8)
Mixed Race	0 (0)	2 (2.3)
Abnormal semen parameters		
Single abnormal parameter		
Sperm concentration ≤ 15 million/mL	4 (4.7)	5 (5.8)
Total motility ≤ 40%	9 (10.6)	10 (11.6)
Normal morphology ^#^ ≤ 4%	33 (38.8)	29 (33.7)
>1 abnormal parameters	39 (45.9)	42 (48.8)
Fathered a prior pregnancy ^^^		
Yes	25 (29.4)	38 (44.2)
No	60 (70.6)	48 (55.8)
Prior infertility treatment and/or surgery		
Yes	25 (29.4)	24 (27.9)
No	60 (70.6)	62 (72.1)
Duration of infertility (months)	24.0	24.0
	(18.0, 48.0)	(15.0, 36.0)
	*n* = 81	*n* = 83
History of smoking		
Never	54 (63.5)	47 (54.7)
Current	8 (9.4)	11 (12.8)
Former	23 (27.1)	28 (32.6)
History of alcohol use		
Never	6 (7.1)	4 (4.7)
Current (in the past year)	72 (84.7)	81 (94.2)
Former (not in the past year)	7 (8.2)	1 (12)

^#^ WHO 5th criteria. ^ *p* < 0.05, Wilcoxon’s rank-sum test was used for the continuous variables, and Chi-square or Fisher’s exact test was used for categorical variables. Wilcoxon’s rank-sum test was used to test the distributional difference, instead of mean or median of the two groups.

**Table 2 antioxidants-12-01626-t002:** Advantages and disadvantages of commonly used techniques to measure seminal oxidative stress. Reproduced with permission from Agarwal, A., Ther Adv Urol. 2016. (Table 1).

Assay	Advantages	Disadvantages
OS via chemiluminescence	Chemiluminescence is robust High sensitivity and specificity Luminol measures global ROS levels—both extracellular and intracellular (superoxide anion, hydrogen peroxide, hydroxyl radical)	Time-consuming method Requires large and expensive equipment Variables such as semen age, volume, repeated centrifugation, temperature control, and background luminescence may interfere with measurement
TAC	Rapid colorimetric method Measures total antioxidants in seminal plasma	Does not measure enzymatic antioxidants Length of the inhibition time is a critical aspect of the test Requires expensive microplate readers
ROS-TAC score	Better predictor compared with ROS and TAC alone	Requires statistical modeling Not a direct measure of ROS or TAC, rather a prediction of oxidative stress
MDA (TBARS adduct by colorimetry or fluoroscopy)	Measures lipid peroxidation Detects MDA-TBA adduct by colorimetry or fluoroscopy	Rigorous controls required Non-specific test providing post hoc measure only
ORP	Provides redox balance in real time Measures all known and unknown oxidants and antioxidants Less time-consuming and requires less expertise Can be measured in semen and seminal plasma, including frozen specimens	Affected by viscosity of the sample

MDA, malondialdehyde; ROS, reactive oxygen species; TAC, total antioxidant capacity; TBARS, thiobarbituric acid reactive substances.

## Data Availability

Not applicable.

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
