# Peer review of "On Males, Antioxidants and Infertility (MOXI): Certitudes, Uncertainties and Trends"

_antioxidants, 2023, doi:10.3390/antiox12081626_

Round 1

Reviewer 1 Report

The review is well organiZed and comprehensive. However I have one remark which might have improved on the scientific context of the review. The authors do not mention about nitric oxide and nitric oxide synthases although their implement in fertility, potence and sexual activity of males is important. NO may reveal a pro- or antioxidant activity in male  fertility or infertility. Its contribution is complex and worth description in the review.

In my opinion English does not demand any particular consideration.

Author Response

It is true, as reviewer #1 says, that the NO and the synthases are very important in the context of IM. In this review we have focused on ROS, as they are the most studied substances and their enzymes, as the text says on page 3 (section 2.1) from lines 119 to 125.

… “most of the articles published over the last 10 years show a strong correlation between elevated reactive oxygen species (ROS) levels in semen and worse reproductive outcomes. Therefore, another certainty is that oxidative stress (OS) is currently the most widely accepted mechanism as a key factor in explaining idiopathic MI cases, [14] through phenomena such as mitochondrial dysfunction, lipid peroxidation, DNA damage and fragmentation and finally, sperm apoptosis [15, 16]”.

However, it is proposed to include on page 2, at the end of the sentence from lines 88 to 91, the following reference (in yellow):

“A fact to highlight is the influence of oxidative stress (OS), which plays an independent role in the etiology of MI, with 30 % to 80 % of infertile men having elevated seminal reactive oxygen species (ROS) and nitrogen species (RNS) levels [5,11].

[11]  Dutta S., Sengupta P., Das S., Slama P.,  Roychoudhury S., Reactive Nitrogen Species and Male Reproduction: Physiological and Pathological Aspects Int J Mol Sci. 2022 12;23(18):10574. doi: 10.3390/ijms231810574.

Others minor changes have been incluided in the text.

Reviewer 2 Report

The manuscript of Gomez et al. aims to provide a comprehensive review concerning current knowledge relating to the role and therapeutic relevance of oxidants in male infertility (MI). To achieve this goal, the Author divided the manuscript into 5 major chapters divided into several subchapters. The “Introduction” describes the definition of male infertility and its diagnosis. The first chapter is dedicated to the known “certainties” linking male infertility to oxidants in three subchapters: “Oxidative Stress and idiopathic male infertility: MOSI”, “Antioxidants, as an alternative in the treatment of idiopathic MI: MOXI” and “Other Antioxidants and their role as Biomarkers”. The second chapter describes the opposite point of view: “Uncertainties in MI” followed by a summary of the currently applied omics methods to evaluate seminal oxidative stress entitled “TRENDS around MOXI”.  

The manuscript is accompanied by 3 illustrative Figures and 2 Tables reproduced from other publications (with permission). Authors cite 85 references.  The manuscript is well written, easy-to-follow and provides a balanced description of diverse opposing opinions in the field.

The topic fits the scope of the journal “Antioxidants” and is of interest for its readers.

This reviewer would ask the Authors to provide one original figure after the “Conclusion” chapter summarizing their view of the oxidants in the development of MI and their therapeutic potential in the future. 

Minor point: Part of the last sentence of the “Abstract” is in different fonts than the rest. 

Author Response

The authors have included a “graphical abstract” that may represent in a plausible way the indications of review #2.

The font in abstract and in the text have been changed.
